

# *In vitro* gastrointestinal digestion study of a novel bio-tofu with special emphasis on the impact of microbial transglutaminase

Guangliang Xing[1], Xin Rui[1], Mei Jiang[1,2], Yu Xiao[1], Ying Guan[1], Dan Wang[1] and Mingsheng Dong[1]

[1] College of Food Science and Technology, Nanjing Agricultural University, Nanjing, P. R. China
[2] Huai'an Academy of Nanjing Agricultural University, Huai'an, P. R. China

## ABSTRACT

We have developed a novel bio-tofu, made from mixed soy and cow milk (MSCM), using *Lactobacillus helveticus* MB2-1 and *Lactobacillus plantarum* B1-6 incorporated with microbial transglutaminase (MTGase) as coagulant. MTGase was added to improve the textural properties and suit for cooking. However, the effect of MTGase on the digestion of mixed-protein fermented by lactic acid bacteria was unclear. This study aimed at evaluating the effect of MTGase on protein digestion of bio-tofu under simulated gastrointestinal digestion condition. The results showed that addition of MTGase could affect the particle size distribution, degree of hydrolysis, the content of soluble proteins and free amino acids. Based on the electrophoresis data, MTGase addition enhanced protein polymerization. During gastric and intestinal digestion process, proteins from bio-tofu were degraded into low molecular mass peptides. Our results suggested that incorporation of MTGase could lead to enzymatic modification of proteins of bio-tofu which may help in controlling energy intake and decrease the chance of food allergy.

## INTRODUCTION

Tofu (soybean curd) is a gel-like food that widely consumed in Asian counties. Conventional tofu is made by coagulating heated soymilk with salt coagulants, like magnesium chloride and calcium sulfate, followed by moulding and pressing the curd to draw the whey. In addition, microbial transglutaminase (MTGase) and glucono-$\delta$-lactone (GDL) have also been used to prepare tofu over the last two decades. Different kinds of coagulants can influence the yield and quality of the final products; for example, the texture of tofu coagulated by GDL and $CaSO_4$ was smoother, while the texture of tofu coagulated by $MgCl_2$ was harder (*Hou et al., 2016*). Also, the addition of MTGase can help maintain the smooth texture of tofu (*Yokoyama, Nio & Kikuchi, 2004*). However, as a single protein

Corresponding author
Mingsheng Dong,
dongms@njau.edu.cn

source of soymilk has not satisfied peoples' nutritional needs, the production of mixed protein matrix is an area of great potential for future development.

Composite gels containing casein (the main cow milk proteins) and soy proteins are possible to be obtained according to the previous studies (*Grygorczyk et al., 2014*; *Lin, Hill & Corredig, 2012*). Formulations containing both soymilk and cow milk proteins provide additional health benefits and also show great potential for new category of food products. The present work focused on a novel soymilk and cow milk mixed tofu, named as bio-tofu, by means of lactic acid bacteria (LAB) incorporated with microbial transglutaminase (MTGase) instead of bittern as coagulant. MTGase (EC 2.3.2.13) is an enzyme that catalyzes the transfer reaction between many proteins by crosslinking of the amino acid residues of protein bound glutamine and lysine (*Hsieh et al., 2014*). Soy proteins and caseins are known to be good substrates for MTGase. Enzymatic modification of proteins by MTGase provides protein to distribute more homogeneously and evenly in network which increases the gel stability and in turn affects the techno-functional properties. Thus, there are numerous health benefits associated with the consumption of bio-tofu containing both proteins and probiotic, and such gels would deliver the health benefits of both dairy and soy products.

It turned out that it is an effective technique of MTGase cross-linking which can be used to improve surface hydrophobic and mechanical properties of protein films, i.e., gelation, emulsification, viscosity and foaming (*Romano et al., 2016*). Particularly, MTGase used in food stuff may modify the immunogenicity of food proteins, such as soy proteins (*Babiker et al., 1998*), peanut proteins (*Clare, Gharst & Sanders, 2007*) and fermented milk beverages (*Wróblewska et al., 2013*). However, the resistant ability of food proteins to the gastrointestinal enzymes is an important factor to take into account which is related to immunological assays (*Villas-Boas et al., 2012*). In some cases the MTGase-catalyzed reaction can affect the stability of proteins with respect to their bioaccessibility (*Rui et al., 2016*), digestibility and allergenicity (*Stanic et al., 2010*; *Tang et al., 2008*). Moreover, *Monogioudi et al. (2011)* reported that none crosslinked $\beta$-casein was less resistant to pepsin digestion when compared to cross-linking of $\beta$-casein by MTGase. According to these findings mentioned above, the novel food structures with improved properties such as controlled energy intake, good satiety and digestibility may develop rapidly in the future. Thus, enzymatic modification of proteins by MTGase could lead to firmer matrices that are digested to a lower extent which may help in controlling energy intake.

Although the study of milk and soy proteins have been conducted, little information is available on the properties of the mixed gels, expecially about the interactions between soymilk and cow milk proteins during acid- (fermented by lactic acid bacteria) and MTGase-induced aggregation. Because of MTGase was used in the present study, it's necessary to determine the digestibility and stability of mixed-proteins through gastrointestinal digestion. The objective of this work was to determine the changes of proteins that take place on addition of MTGase or not to mixed soymilk and cow milk fermented by *Lactobacillus helveticus* MB2-1 and *Lactobacillus plantarum* B1-6 at different points and evaluate the protein degradation and release of peptides/amino acids of bio-tofu by an *in vitro* gastrointestinal digestion (GIS) model.

## MATERIALS AND METHODS

### Materials

$\alpha$-Amylase (A1031), pepsin (P7000), bile acid (B8631), pancreatin (P3292) were purchased from Sigma-Aldrich Co. (St. Louis, MO, USA). Molecular mass standard (15–150 kDa) was purchased from Sangon Biotech Co. (Shanghai, China). All other regents used were of analytical grade.

### Preparation of bio-tofu

*L. helveticus* MB2-1 and *L. plantarum* B1-6 were isolated in our lab from Sayram ropy fermented milk and Kirgiz boza respectively, which are all traditional food collected from Xinjiang province of China (*Li et al., 2012*; *Wu et al., 2015*). *L. helveticus* MB2-1 had been deposited in GenBank database with accession number CP011386 and *L. plantarum* B1-6 was given gene accession number KM200717. *L. plantarum* B1-6 strains were activated twice in de Man-Rogosa and Sharp broth (MRS, pH 6.2 ± 0.2, Oxoid-CM0361, Unipath, Basingstoke, UK) at 37 °C for 24 h and 16 h prior to use. *L. helveticus* MB2-1 strains were propagated for two successive transfers in cow milk (total fat-3.7 g, total carbohydrate-4.8 g, protein-3.0 g per 100 ml; Mengniu Dairy Group, China) at 42 °C for 24 h and 16 h prior to use in experimental trials.

The step-by-step preparation of bio-tofu is shown in Fig. 1. Briefly, soaked soybeans were ground twice at high speed for 5 min by a homogenizer (BE601AB; Midea, Guangdong, China) with hot water (90–95 °C) at a bean:water ratio of 1:9 (m/v). After grinding, a 200-mesh screen cloth was used to remove the okara, which was mainly insoluble fibre. Then the raw soymilk was obtained. Subsequently, this soymilk was mixed with cow milk for a final protein content of 1.6% milk protein and 1.6% soy protein in the mixture. The resultant mixed-liquid was heated in a pan, with constant stirring at 100 °C, on a Joyoung induction cooker (C21-QPAD1; Joyoung, Hangzhou, China) for about 5 min. After cooling to incubation temperature (38–40 °C), the mixed soy and cow milk (MSCM) were inoculated with 4.0% (v/v) of a mixed culture (1:1 v/v) of *L. plantarum* B1-6 and *L. helveticus* MB2-1. Subsequently, MTGase (110 U/g) was added into the solution with stirring at concentration of 3.0 U/g (based on the content of protein). Inoculated MSCM with adding MTGase were poured into many 100 mL sterile cups and incubated at 38 °C for 5 h to make bio-tofu (BT-with MTGase). In the production of samples, the same procedure mentioned above were followed but without adding MTGase to form another kind of curd (BT-without MTGase). After incubation, all samples were cooled down and stored at 4 °C for analysis.

### *In vitro* GIS digestion

The *in vitro* buccal, gastric and intestinal juice was prepared as described by *Shim et al. (2010)* with some modifications to mimic human digestion. The whole GIS digestion steps were carried out sequentially in beakers placed in a shaking water bath (SWB series, Biobase, Shandong, China) at 37 °C which is shown in Fig. 2. Several GIS digestions of both BT-with MTGase and BT-without MTGase were performed to obtain digested samples at different times during the digestion process. The digestion times studied were as follows:

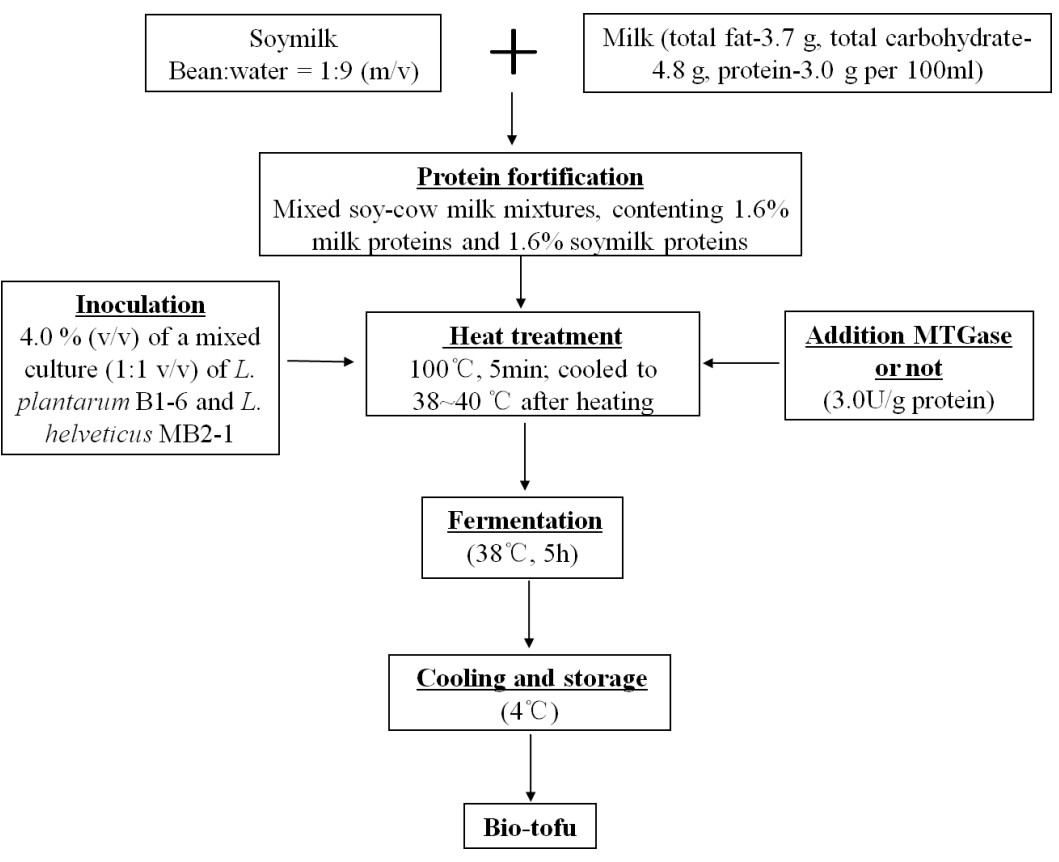

**Figure 1** Schematic diagram of making bio-tofu.

before digestion (P0); at the end of buccal phase (3 min, P1); after 1, 5, 25 and 60 min of gastric phase (P2-1, P2-5, P2-25, P2-60) and after 1, 5, 30, 120 min of duodenal phase (P3-1, P3-5, P3-30, P3-120). After each digestion time, each sample was heat-treated at 95 °C for 5 min to stop the enzymatic digestion and then centrifuged (CT15RT, 10,000× g, 20 min, 4 °C) except those prepared for particle size distribution. The supernatants were kept frozen at −20 °C until use.

## Particle-size distribution

The particle-size distribution of the digestion samples was determined using a Mastersizer 3000 (Malvern, Southborough, MA, USA). A particle refractive index of 1.39 was used for caseins, 1.46 for soy proteins dispersions and 1.42 for the mixed systems (*Lin, Hill & Corredig, 2012*). The refractive index of the dispersing phase (water) was 1.33. The volume-weighted mean diameter D [4,3] was characterized as the size of gel particles. D [v,0.90] was also reported to describe the diameter below which 90% of the volume of particles were found. The analysis was conducted in triplicates.

## Total soluble protein content of digested samples

Soluble protein in the supernatants from digested BT-with MTGase and BT-without MTGase were quantified with the Bradford method (*Bradford, 1976*) using bovine serum

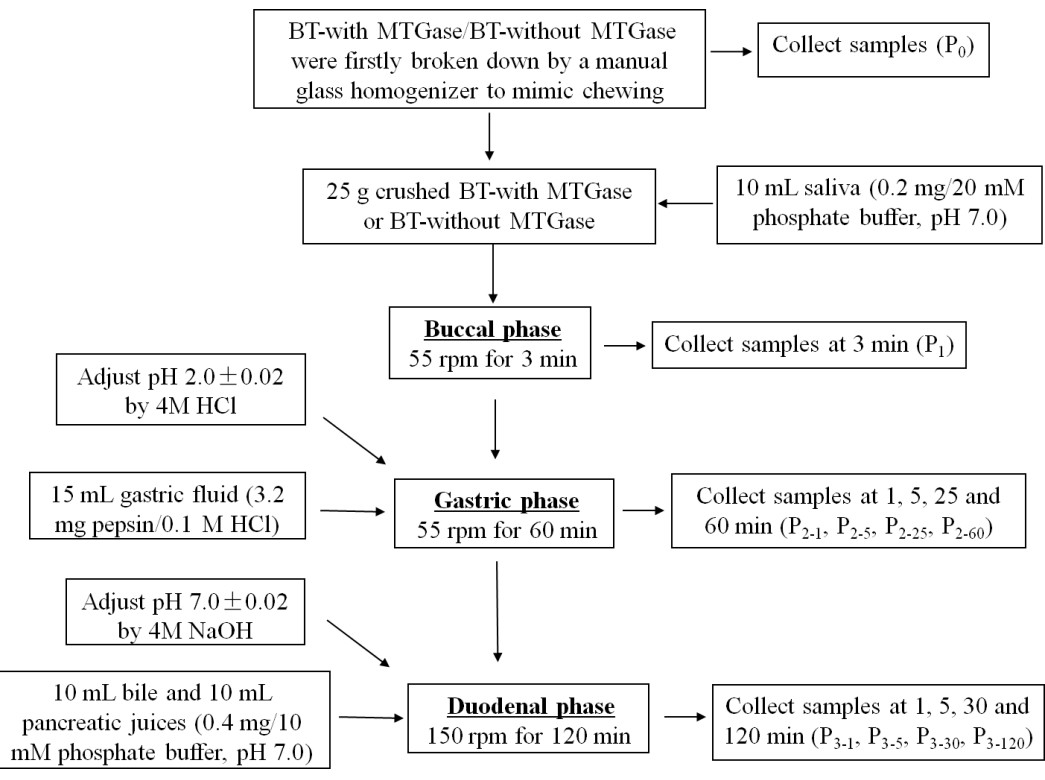

**Figure 2** The process of *in vitro* gastrointestinal simulated digestion (GIS).

albumin (Sigma) as the standard. A microplatereader (VersaMax ELISA Microplate Reader; Molecular Devices, Sunnyvale, CA, USA) was used to determine the concentration spectrophotometrically (595 nm). The soluble protein content in the supernatant of digestion mixtures was expressed as milligram protein in per milliliter GIS digestion fluid (mg/mL). Each sample was analyzed in triplicates.

## Electrophoresis

Aliquots taken at different digestion time points were evaluated by sodium dodecyl sulfate-polyacrylamide gel electrophoresis (SDS-PAGE), which was conducted at a presence of 5% $\beta$-mercaptoethanol by using a 12% polyacrylamide gel. The gels were stained with 0.1% (w/v) Coomassie brilliant blue R-250 (Sigma) at room temperature. SDS-PAGE was conducted on a Bio-Rad Miniprotein 3 unit (Bio-Rad Laboratories, Inc., Hercules, CA, USA) with voltage 60 V for stacking gel, and followed by 120 V for separating gel. The molecular weight values of the protein fractions were estimated using broad-range Protein Marker (15–150 kDa). The protein-stained bands were scanned with Image Scanner III (GE Healthcare Biosciences, Uppsala, Sweden) and analyzed using Quantity One software, version 4.6.2 (Bio-Rad Laboratories, Inc., Hercules, CA, USA).

## Degree of hydrolysis

The determination of the degree of hydrolysis (DH) during *in vitro* GI digestion was carried out using the o-phthaldialdehyde (OPA) method by *Church et al. (1983)*. Briefly, 400 µL of
the supernatant was added to 3 mL of OPA reagent and left at room temperature for 2 min. After 2 min, the absorbance of the solution was measured using a spectrophotometer at 340 nm. Serine was used as a standard, and measurements were realized in duplicates.

### Protein digestibility assay—pH drop method

The pH drop method was used to determine the rate of digestibility of the BT-with MTGase and BT-without MTGase according to previous study (*Hsu et al., 1977*). Digestibility of each sample was calculated based on the change in pH after 10 min of digestion (X) using the equation: Digestibility = 210.46–18.10X. The analysis was conducted in triplicates.

### Free amino acid determination

Free amino acid determination was analyzed according to a previous procedure (*Aro et al., 2010*). Samples collected at varied digestion times were mixed with 4% trichloroacetic acid (TCA) at volume ratio of 1:1 and then incubated at 37 °C for 30 min. The mixture was subsequently filtered using a 7 cm Whatman filter paper disc, and the filtrates collected were further applied to 0.45 $\mu$m filters. 20 $\mu$l of the sample were subjected to a fully automated amino acid analyser HITACHI L-8900 (Hitachi Ltd., Tokyo, Japan).

### Statistical analysis

The data were subjected to independent student's *t*-test to determine the significant difference between means at $P < 0.05$ level using IBM SPSS Statistics.

## RESULTS AND DISCUSSION

### Particle size distribution

The particle size distribution of the BT-with MTGase and BT-without MTGase before and after *in vitro* gastrointestinal digestion are shown in Fig. 3. The figures clearly suggest a bimodal distribution for BT-with MTGase with a size range from 5.92 to 454 $\mu$m and BT-without MTGase with a size range from 4.58 to 352 $\mu$m (Fig. 3A). The particle size (D[4,3]) in BT-with MTGase and BT-without MTGase were significantly different ($P < 0.05$). Besides, BT-with MTGase had larger value of D[v,0.90] (236.33 $\pm$ 2.31 $\mu$m) which indicated a tight and compact structure of proteins obtained through cross-linking by MTGase. Nevertheless, the addition of simulated saliva fluid to the two samples led to the particle size distribution increased remarkably (Fig. 3B). This might due to proteins precipitation during the heat-treatment of stopping the $\alpha$-Amylase digestion process. The subsequent *in vitro* gastric digestion (P2) altered larger protein particles to smaller ones which appeared as a result of the breakdown by pepsin (Fig. 3C). Meanwhile, a decrease of the volume mean D[4,3] diameter was also detected. The phenomenon was particularly observed for BT-with MTGase, as indicated by D[4,3] value reduced from 163.00 $\pm$ 6.24 $\mu$m (P1) to 24.60 $\pm$ 2.05 $\mu$m (P2) (Figs. 3B and 3C ). After 120 min of duodenal phase, the size shift rate of these two samples was slower, but a higher level of much smaller particles (<10 $\mu$m) was generated (Fig. 3D). BT-with MTGase contained larger D[4,3] and D[v,90] values compared to BT-without MTGase. This observation also showed that large protein particles were harder to break down by the digestive enzymatic which were assembled by
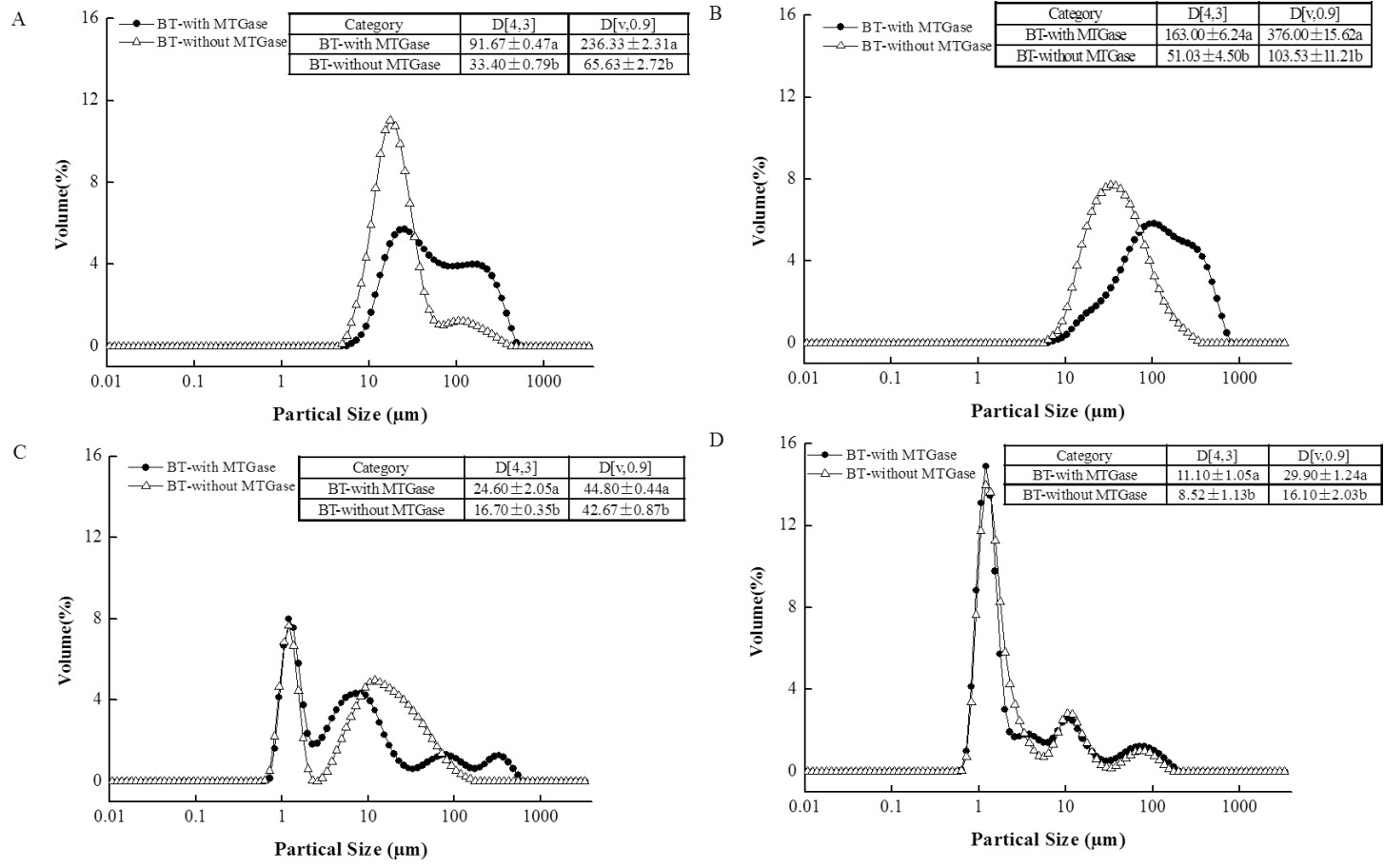

**Figure 3** Particle size distribution of the mixed soy-cow protein samples (A) before the GIS digestion, (B) after buccal digestion, (C) after gastric digestion, (D) after intestinal digestion. Different patterns represented bio-tofu with MTGase (BT-with MTGase −●−), and bio-tofu without MT-Gase (BT-without MTGase −△−). Data are expressed as mean ± SD from triplicate experiments. Different letters within the same column indicate significant difference ($P < 0.05$).

MTGase. The MTGase cross-linking influenced the gastric and duodenal digestibility of BT-with MTGase, making the protein digest slower than BT-without MTGase.

## Protein degradation under simulated gastrointestinal digestion condition

Figure 4 shows the soluble protein content in the supernatants obtained from digestion of the mixed soy-cow protein samples. Due to the presence of digestive enzymes, the amounts of soluble proteins changed obviously. Before digestion (P0), low amounts of soluble proteins were detected for both BT-with MTGase and BT-without MTGase. No change of solubilization of proteins in BT-without MTGase at the buccal digestion (P1) phase but a slight increase in that of BT-with MTGase. During the subsequent gastric digestion, the protein solubility in both samples were increased drastically. The amounts of soluble proteins of BT-with MTGase and BT-without MTGase increased sharply to 2.35–3.11 mg/mL and 5.03–5.56 mg/mL respectively, which could be explained by the hydrolytic action of pepsin. The organized protein network collapsed quickly and soluble

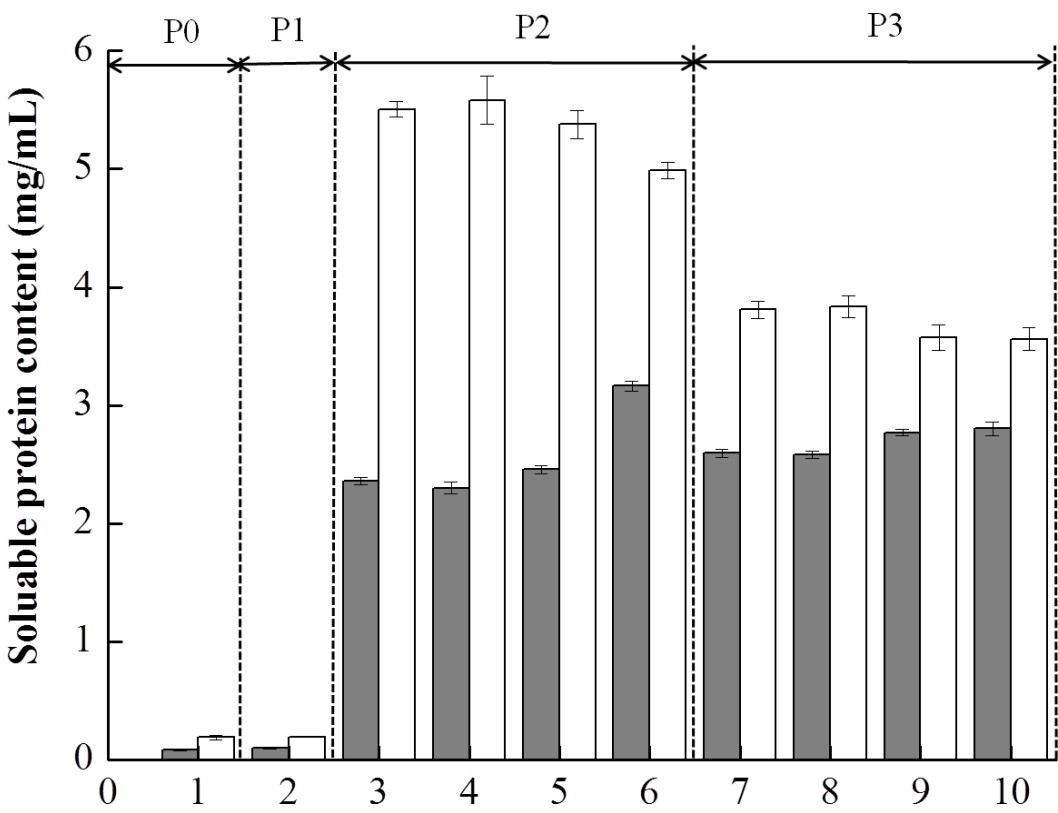

**Figure 4** **Soluble protein content (mg/mL) of the mixed soy-cow protein samples subjected to *in vitro* gastrointestinal simulated digestion (GIS): 1-before the GIS (P0); 2-after buccal digestion (P1); 3,4,5,6- represented samples taken at 1 min, 5 min, 25 min and 60 min of gastric digestion (P2); 7, 8, 9, 10- represented samples taken at 1 min, 5 min, 30 min and 120 min of intestinal digestion (P3).** Different patterns represented the bio-tofu with MTGase (BT-with MTGase, ■) and bio-tofu without MTGase (BT-without MTGase, □).

proteins released dramatically due to the gastric environment, including the acidic pH, the presence of pepsin and continuous mechanical shaking. Those results were in accordance with the studies of *Rinaldi et al. (2014)* and *Rinaldi et al. (2015)*. Protein solubility peaked at 5 min and 60 min of gastric digestion for BT-without MTGase and BT-with MTGase, respectively. During the duodenal digestion phase (P3), the pattern changed and resulted in the decrease of the soluble protein content for both samples. The formation of peptides and amino acids during P3 had lower molecular masses which were undetected by Bradford assay, thus underestimated results were observed. Similar observations were reported by *Rioux & Turgeon (2012)* to determine the effects of milk protein composition on *in vitro* digestion. Overall, the dilution related to the addition of the duodenal solution, the pH changes and the rapid hydrolysis resulted in similar decrease values for both samples.

The amounts of soluble proteins changed as a result of the hydrolysis initiated by proteolytic enzymes in the process of digestion. However, the addition of MTGase had remarkable influences on the *in vitro* digestibility of protein. Similar observation was also reported which showed that the presence of MTGase would retard the rate of digestion in an *in vitro* gastric digestion (*Macierzanka et al., 2012*), even through with different

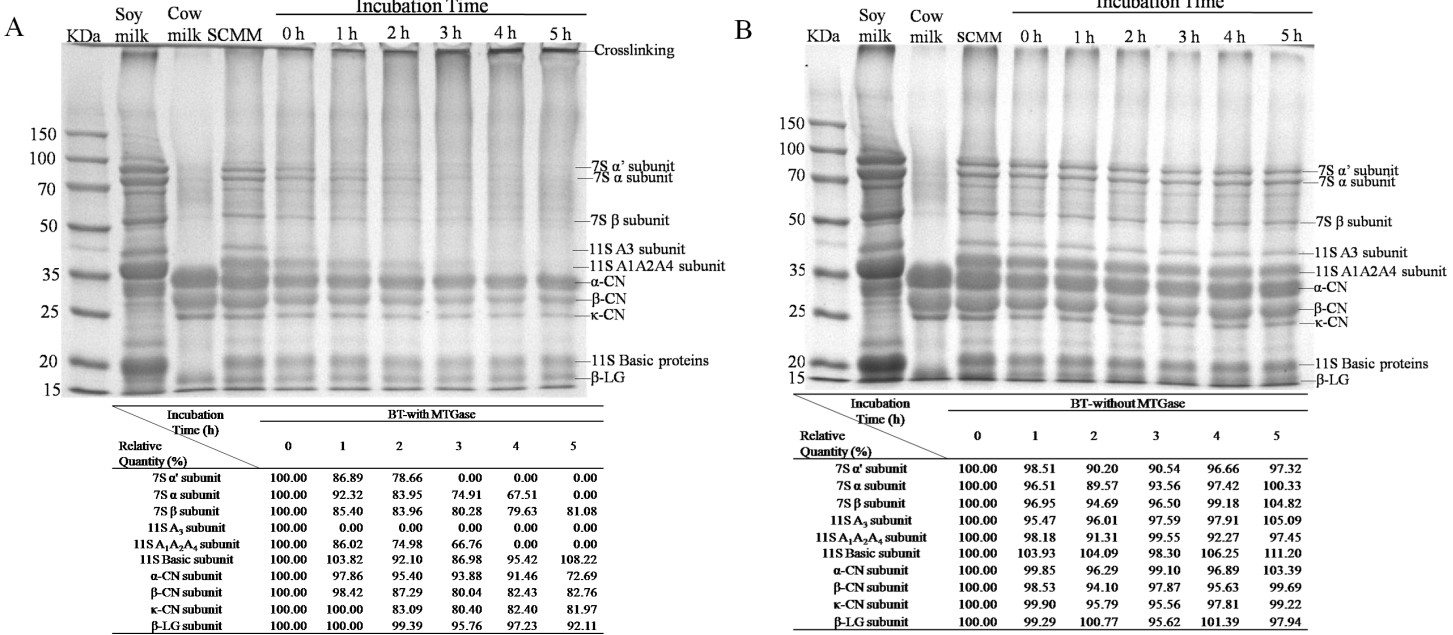

**Figure 5** Sodium dodecyl sulfate polyacrylamide gel electrophoresis (SDS-PAGE) profiles of the soy-cow milk mixtures fermented with addition of MTGase (A) or not (B) for different periods from 0 h to 5 h. 7S $\alpha'$-, 7S $\alpha$- and 7S $\beta$-: subunits of $\beta$-conglycinin; 11S A3, 11S A1A2A4 and 11S Basic: acidic and basic subunits of glycinin; $\alpha$-CN, $\beta$-CN and $\kappa$-CN of casein; $\beta$-LG of whey protein. Relative quantity (%) of every protein band was found within the table.

concentrations (*Rui et al., 2016*). This was probably due to the formation of cross-linking bonds among proteins made it more difficult for enzymes in the digestive fluid to attack, which led to much fewer proteins digested. Therefore, the food microstructure had an impact on the solubilization of proteins or their hydrolysis during the GIS digestion. In addition, these results could be exploited in novel food matrices that, because of its lower digestibility by the adding MTGase, could provide commercial products with controlled energy intake (*Romano et al., 2016*).

### *In vitro protein digestion determined by SDS-PAGE*

The MSCM with addition of MTGase or not were incubated at 38 °C for 0, 1, 2, 3, 4, and 5 h prior to analysis by SDS-PAGE (Figs. 5A and 5B). The protein profiles were analyzed by Quantity One software. The results of relative quantity were found within the table below the Figs. 5A and 5B.

It is well known that the two major soymilk proteins are $\beta$-Conglycinin (7S) and glycinin (11S). $\beta$-Conglycinin is a trimer formed from various combinations of the three subunits ($\alpha'$, $\alpha$, and $\beta$) and 11S is a hexameric protein consisting of six subunits (*Aguirre et al., 2014*). The major proteins of cow's milk are casein and whey. Five different forms were contained in casein ($\alpha_{s1}$-CN, $\alpha_{s2}$-CN, $\beta$-CN, $\gamma$-CN, and $\kappa$-CN) and whey has two subunits ($\alpha$-lactalbumin ($\alpha$-LA) and $\beta$-lactoglobulin ($\beta$-LG)) (*Chen et al., 2014*). No significant changes were observed in BT-without MTGase after a 5-h incubation (Fig. 5B), indicating that proteolysis during fermentation by *L. helveticus* MB2-1 and *L. plantarum* B1-6 were not intense. Besides, protein polymerization did not occur in the absence

of MTGase. Whereas cross-links between soymilk and cow milk proteins were seen to form during fermentation in the presence of MTGase (Fig. 5A). A concurrent increase in high-molecular-mass polymer(s) was/were observed which correlated positively with increasing incubation time. Moreover, the SDS–PAGE profiles of the BT-with MTGase showed a progressively decrease in the intensities of both soymilk protein (7S and 11S) and cow milk protein (casein and $\beta$-lactoglobulin) bands. In particular, the intensity of the $\alpha'$, $\alpha$, $\beta$ subunits of $\beta$-conglycinin (7S) dropped rapidly at 0 h incubation. Acidic subunits of glycinin (11S) were also depleted, followed by the basic subunits and cow milk subunits, which were the least reactive. It was noteworthy that the majority of the 11S A3 subunit disappeared after a 1-h incubation period (Fig. 5A), showing that the enzymatic reactivity for the 11S A3 subunit was much higher than that for other 11S acidic proteins. This observation was in agreement with the results reported by *Yasir et al. (2007)*. Most likely, the 11S A3 subunit is located on the surface of the 11S molecule thus is the most accessible and reactive monomer that can be attacked by MTGase more easily.

SDS-PAGE separated the major cow milk proteins, including the $\alpha$-CN, $\beta$-CN, $\kappa$-CN and $\beta$-LG as shown in Fig. 5. The molecular weight of each protein was 33.4, 27.3, 24.7 and 17.0 kDa, respectively. The bands corresponding to casein fractions became less intense as the incubation time increased (Fig. 5A). Compared with casein, the $\beta$-LG acted less effectively. It had been reported that whey proteins had lower tendency to form cross-link reaction than caseins by MTGase (*Şanlı et al., 2011*). In general, the native fold of the proteins (*Tang et al., 2006*) and the amino acid sequence specificity of MTGase (*Kamiya et al., 2003*) related to the differences in reactivities of the subunits.

The identification of the soluble proteins found in the supernatant during *in vitro* GIS were further visualized as shown in Figs. 6A and 6B. A limited number of bands with molecular mass (MM) of 38 and 30 kDa (band 1 and 2, Fig. 6A) were detected in the supernatant of BT-with MTGase at P0 phase which was in agreement with the earlier finding of low soluble protein content. After buccal digestion (lane P1, Fig. 6A), the pattern was unchanged since there were no proteases in buccal fluids. Upon digestion of pepsin (lane P2, Fig. 6A), revealing the existence of resolved protein bands ranging from 22–33 kDa (band 3), $\beta$-LG at 17 kDa (band 4). Band 3 and band 4 were stable at all time points of pepsin digestion. Band 3 appeared to be partially hydrolyzed while still visible at P2-60. In terms of band 4, $\beta$-LG was resistant to enzymatic digestion, particularly to pepsin, which was interrelated to its complex structure in acidic pH (*Chicón et al., 2008*). During the duodenal digestion phase (lane P3, Fig. 6A), band 3 (21–30 kDa) was stable while band 4 was undetected at 1 min time point compared to the 17 kDa band which could be seen in the lane P2, as in findings by *Do, Williams & Toomer (2016)*. Meanwhile, an indeterminate smear (lane P3, Fig. 6A) was formed at different time point by pancreatin, indicating abundant heterogeneity of the molecular masses of the digested protein fragments.

The SDS-PAGE patterns of BT-without MTGase through the whole digestion are shown in Fig. 6B. AS shown in lane P2 (Fig. 6B), protein bands of 76 kDa (band 1), 56 kDa (band 2) were not exsit in lane P2 of Fig. 6A, but these two samples had the same MM range from 22 to 33 kDa (band 3). Band 3 (lane P2 and P3, Fig. 6B) was stable at all time points of gastric and duodenal digestion. It is thought that in order to reach the immune system

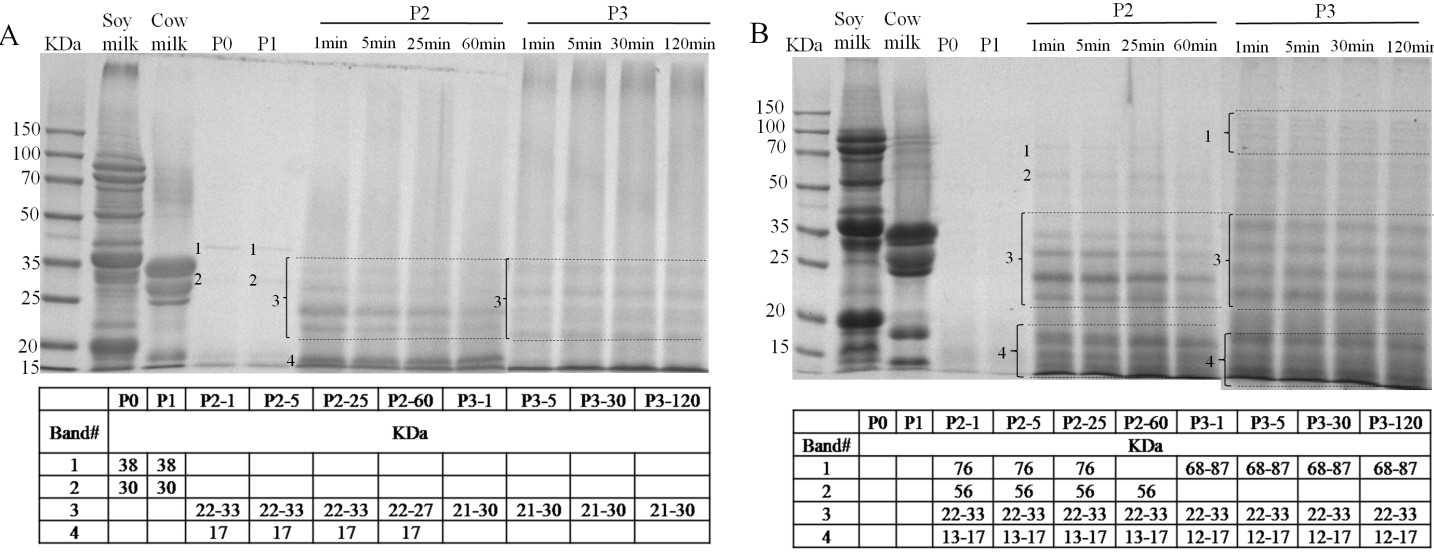

**Figure 6** **Sodium dodecyl sulfate polyacrylamide gel electrophoresis (SDS-PAGE) analysis of digested samples before the GIS digestion (P0), during buccal (P1), gastric (1 min: P2-1, 5 min: P2-5, 25 min: P2-25, 60 min: P2-60) and duodenal (1 min: P3-1, 5 min: P3-5, 30 min: P3-30, 120 min: P3-120) phases of *in vitro* GIS digestion.** (A) bio-tofu with MTGase (BT-with MTGase); (B) bio-tofu without MTGase (BT-without MT-Gase). Numbered protein bands correspond to values of molecular mass (kDa) found within the table.

of the intestine, an allergen is able to survive harsh acidic and proteolytic environment of the stomach, or to share the epitopes with common aeroallergens (*Mills et al., 2004*). Thus, the more proteins bands were detected through the gastrointestinal digestion, the bigger chance of being an allergen.

In general, after gastric digestion, intensive bands could be observed in the sample of BT-with MTGase (lane P2, Fig. 6A), but weaker than BT-without MTGase (lane P2, Fig. 6B). Besides, gastric digestion allowed the presences of more visible bands in BT-without MTGase than BT-with MTGase (i.e., the band 1 and 2 in lane P2 Fig. 6B couldn't be observed in lane P2 Fig. 6A). Under the hydrolysis of pepsin, some new bands were detected which were not existed in the original soy milk and cow milk profile. This pattern maintained through the gastric digestion for these two samples. During the duodenal digestion phase, the patterns were similar throughout digestion but more intensive bands were observed in the low molecular mass region, which might reflect a rapid degradation of protein. Proteins had been degraded further into fragments of lower molecular mass which were not detected by SDS-PAGE. In general, the enzymatic modification of soy and cow milk proteins by MTGase could lead to firmer matrices that were digested to a lower extent which indicated the less chance to induce food allergy.

## Digestibility determined by pH drop methods and the degree of hydrolysis

Table 1 shows the *in vitro* protein digestibility of these two mixed soy-cow protein samples. It was found that BT-without MTGase showed significantly higher value ($P < 0.05$) (74.65 $\pm$ 0.75%) than BT-with MTGase (72.18 $\pm$ 0.58%). The protein digestibility results indicated that BT-without MTGase was much more susceptible to digestive enzymes, whereas strong

**Table 1** *In vitro* digestibility of the bio-tofu with MTGase (BT-with MTGase) and bio-tofu without MTGase (BT-without MTGase). Mean values of digestibility that do not share the same letter (a or b) indicate significant difference ($P < 0.05$). Triplicate samples were measured from duplication.

| Sample | *In vitro* digestibility |
|---|---|
| BT-with MTGase | $72.18 \pm 0.58^b$ |
| BT-without MTGase | $74.65 \pm 0.75^a$ |

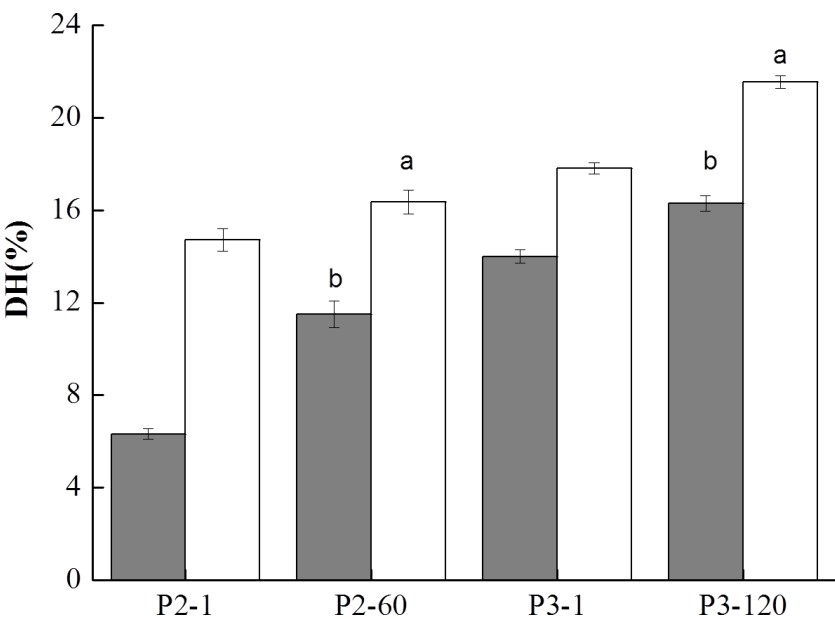

**Figure 7** Degree of hydrolysis (DH) of digested samples at the gastric and duodenal steps for the bio-tofu with MTGase (BT-with MTGase, ■) and bio-tofu without MTGase (BT-without MTGase, □). Values are means ± SD of three independent experiments ($n = 3$). Different letters at the top of the bars indicate significant difference ($P < 0.05$) between dairy samples at the end of each digestion phase.

inter/intra molecular bonds were formed by MTGase-catalyzed cross-linking between soy and milk proteins which hindered digestive enzymatic hydrolysis.

Proteolysis as assessed by the release of free NH3 groups was quantified using OPA assay after 1, 60 min of gastric digestion (P2-1, P2-60) and after 1, 120 min of duodenal digestion (P3-1, P3-120) which are shown in Fig. 7. During GIS digestion, soy proteins and cow milk proteins were degraded by pepsin and pancreatin, resulting in the release of various forms of peptides and free amino groups. As depicted in Fig. 7, initiating of gastric digestion for 1 min (P2-1) resulted in a dramatically elevation of the DH level which indicated many proteins were hydrolyzed into large peptides by pepsin. At the end of gastric digestion (P2-60), the DH of the BT-without MTGase (16.36%) was significantly higher ($P < 0.05$) than BT-with MTGase (11.49%). The DH slightly increased for these two samples at the beginning of duodenal digestion. At the end of the whole digestion, BT-without MTGase still showed dramaticlly higher DH (21.55%) than BT-with MTGase (16.30%) ($P < 0.05$). In general, a fast hydrolysis and an increase in the number of low molecular mass peptides at each digestion step contributed to the increase of the DH level.

**Table 2  Free amino acids contents of bio-tofu with MTGase (BT-with MTGase) and bio-tofu without MTGase (BT-without MTGase) at the end of *in vitro* GIS digestion.**  Results are expressed in milligrams per liter digestion solution.

|  | BT-with MTGase | BT-without MTGase |
| --- | --- | --- |
| Asp | 0.47 | 1.131 |
| Thr | 1.391 | 2.221 |
| Ser | 1.168 | 1.618 |
| Glu | 5.800 | 10.050 |
| Gly | 1.195 | 1.922 |
| Ala | 3.327 | 5.956 |
| Cys | 0.934 | 1.477 |
| Val | 0.286 | 0.207 |
| Met | ND[a] | 0.127 |
| Ile | 0.100 | 0.149 |
| Leu | 2.047 | 3.352 |
| Tyr | 5.098 | 9.537 |
| Phe | 6.519 | 9.774 |
| Lys | 2.328 | 4.697 |
| His | 1.692 | 3.165 |
| Trp | 0.852 | 2.724 |
| Arg | 8.016 | 14.827 |
| Pro | 7.177 | 8.738 |
| Essential amino acids | 13.523 | 23.251 |
| Total | 48.400 | 81.671 |

**Notes.**
[a] ND, not determined

## Amino acids analysis

At the end of *in vitro* GIS digestion, the content of free amino acids (FAA) in the two samples was measured and the results are shown in Table 2. The presence of peptidases in pepsin and pancreatin added in digestion juice resulted in a considerable increase of FAA content in both samples. As shown in Table 2, eighteen amino acids were determined and eight of them were corresponding to essential amino acids (Thr, Val, Met, Ile, Leu, Phe, Lys and Trp). The concentration of total FAA content in BT-without MTGase (81.671 mg/L) was 1.69-fold higher than BT-with MTGase (48.400 mg/L).

At the end of digestion, the most abundant amino acids released (in milligrams per liter) in BT-without MTGase were Arg (14.827), Glu (10.050), Phe (9.774), Tyr (9.537), and Pro (8.738), followed by Ala (5.956), Lys (4.697), Leu (3.352), and His (3.165), together representing 85.83% of total FAA. The content of most essential FAA (Thr, Ile, Phe, Lys, Trp) in BT-with MTGase were much lower than BT-without MTGase, but only a few of them (Val and Leu) were slightly higher.

The content of FAA also reflected the SDS-PAGE patterns obtained in the end of the *in vitro* GIS digestion for the samples. The SDS-PAGE pattern of BT-with MTGase revealed fewer faint bands when compared to BT-without MTGase in the low molecular weight region and this suggested that protein material was intensively hydrolyzed as short peptides

and FAA which were not retained by the gels. At the end of the duodenal phase, the amount of peptides bands appeared in the low molecular weight region of the gels followed the same increasing order than the FAA content: BT-without MTGase >BT-with MTGase. Thus, addition of MTGase to soy-cow milk mixtures might help enhance satiety, control energy intake or lose weight.

## CONCLUSIONS

In the present study we have developed a novel bio-tofu, made from mixed soy and cow milk (MSCM) using *Lactobacillus helveticus* MB2-1 and *Lactobacillus plantarum* B1-6 incorporated with microbial transglutaminase (MTGase) as a coagulant. This kind of bio-tofu was a good source of protein and contained all eight essential amino acids. It was also an excellent source of iron and calcium. In addition, the use of probiotics in bio-tofu was highly beneficial to human health. Investigation was carried out to explore the changes of proteins that take place on addition of MTGase or not to MSCM at different points and evaluate the protein degradation profiles by an *in vitro* gastrointestinal digestion (GIS) model. The protein hydrolysis in the gastric and duodenal digestion phases were different for BT-with MTGase compared to BT-without MTGase as expected. The addition of MTGase seemed to influence the proteins' behavior and then affected the digestibility. On the other hand, food structures with higher satiety effects and lower allerginicity might be produced via enzymatic cross-linking by MTGase. Only an *in vitro* digestibility model was used in the study to evaluate the protein degradation, which might not predict the protein stability *in vivo*. Further investigations are needed to elucidate whether MTGase affects the allergenicity of bio-tofu during digestion in allergic individuals or testes in an animal model.

### Funding

This work was co-financed by the Scientific and Technical Project of Huai'an city, Jiangsu Province (HAC2015020). This research was also supported by Jiangsu Collaborative Innovation Center of Meat Production and Processing, Quality and Safety Control. The funders had no role in study design, data collection and analysis, decision to publish, or preparation of the manuscript.

### Grant Disclosures

The following grant information was disclosed by the authors:
Scientific and Technical Project of Huai'an city, Jiangsu Province: HAC2015020.
Jiangsu Collaborative Innovation Center of Meat Production and Processing, Quality and Safety Control.

### Competing Interests

The authors declare there are no competing interests.

## Author Contributions

- Guangliang Xing conceived and designed the experiments, performed the experiments, analyzed the data, contributed reagents/materials/analysis tools, wrote the paper, prepared figures and/or tables, reviewed drafts of the paper.
- Xin Rui analyzed the data, contributed reagents/materials/analysis tools, wrote the paper, reviewed drafts of the paper.
- Mei Jiang analyzed the data, contributed reagents/materials/analysis tools, this work was co-financed by Scientific and Technical Project of Huai'an city, Jiangsu Province (HAC2015020).
- Yu Xiao contributed reagents/materials/analysis tools.
- Ying Guan and Dan Wang performed the experiments.
- Mingsheng Dong conceived and designed the experiments, analyzed the data, contributed reagents/materials/analysis tools, wrote the paper, reviewed drafts of the paper.

## Data Availability

The raw data has been supplied as a Data S1.

## Supplemental Information

Supplemental information for this article can be found online at http://dx.doi.org/10.7717/peerj.2754#supplemental-information.

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
