# Peer review of "In vitro gastrointestinal digestion study of a novel bio-tofu with special emphasis on the impact of microbial transglutaminase"

_PeerJ, doi:10.7717/peerj.2754_

## Round 0.1 · original submission · Minor Revisions

· Academic Editor

Minor Revisions

Overall, the reviews are favorable but revisions are needed before the manuscript can be considered further. I urge the authors to address the concerns raised.

Reviewer 1 ·

Basic reporting

This is an interesting study to describe the effect of microbial transglutaminase-added bio-tofu on in-vitro digestibility. The overall manuscript has been presented in a scientific way, with novel findings and concise discussions, thus merits for suitable publication in PeerJ. All tables and figures have been illustrated clearly.

Experimental design

The experimental design was clear, appropriate and allows future replication.

Remark: I would rather suggest the author to change ANOVA analysis to independent student t test for the direct comparison between with and without bio-tofu (only 2 groups). As such, Duncan's post hoc is not needed.

Validity of the findings

I believe that the findings are valid and up to standard.

Remark: The authors are suggested to add more discussions on the beneficial effects of bio-tofu in human health. Also please include the limitation of your work (ie: only in-vitro study to evaluate digestibility).

Additional comments

The overall manuscript is merit for publication.

Reviewer 2 ·

Basic reporting

The manuscript described a procedure of making a novel bio-tofu from a combination of soy and cow milk using MTGase as coagulant, and compared in vitro gastrointestinal digestion of the bio-tofu with and without MTGase. The authors reported that addition of MTGase affected protein structure in the tofu with increased particle size and tightness and therefore slowed its digestion. Protein degradation was monitored under the simulated gastrointestinal environment, and protein profiles, degree of hydrolysis and amino acid contents were measured to determine the digestion status of the product.

Experimental design

The experimental design is appropriate and the findings are valid.

Validity of the findings

Fine!

Additional comments

The manuscript is well-written and relatively easy to follow, yet it willl increase readability if you can ask a native English speaker to check the grammar, spelling and word use.
Specific comments:
Line 54: “possible to obtained” should be “possible to be obtained”?
Line 76: “which related to” should be “which is related to”?
Line 256: “which lead to” should be “which led to”?
Line 363: “increasing of the DH” should be “increase of the DH”?
Line 337-340, 363 and 383: please keep consistency in using past tense describing the results.

Reviewer 3 ·

Basic reporting

Language should be further polished.

Figures should be carefully labelled.

Table 1 a, b?

Experimental design

The experiments are well designed, but lack scientific significance.

Validity of the findings

The data is solid and sound

---

## Round 0.2 · accepted · Accept

· Academic Editor

Accept

Concerns have been addressed adequately in the revised version.

Reviewer 1 ·

Basic reporting

The revised version has been improved significantly.

Experimental design

The experimental design is valid and allows future replication.

Validity of the findings

I believe that the findings are valid and up to standard.

Additional comments

The current version is acceptable.

Reviewer 2 ·

Basic reporting

I have reviewed all the files submitted including the letter to the Editors, the responses to the reviewers' comments and the revised manuscript. It seems to me that the authors have addressed all the reviewers' concerns/suggestions. I am satisfied with the revision.

Kind regards,

Guijun

Experimental design

As above.

Validity of the findings

As above.

Additional comments

Well done, authors!